# The Genetic Analyses of French Canadians of Quebec Facilitate the Characterization of New Cancer Predisposing Genes Implicated in Hereditary Breast and/or Ovarian Cancer Syndrome Families

**DOI:** 10.3390/cancers13143406

**Published:** 2021-07-07

**Authors:** Caitlin T. Fierheller, Wejdan M. Alenezi, Patricia N. Tonin

**Affiliations:** 1Department of Human Genetics, McGill University, Montreal, QC H3A 1A1, Canada; caitlin.fierheller@mail.mcgill.ca (C.T.F.); wagdan.alenizy@mail.mcgill.ca (W.M.A.); 2Cancer Research Program, Centre for Translational Biology, The Research Institute of the McGill University Health Centre, Montreal, QC H4A 3J1, Canada; 3Department of Medical Laboratory Technology, Taibah University, Medina 42353, Saudi Arabia; 4Department of Medicine, McGill University, Montreal, QC H3A 1A1, Canada

**Keywords:** French Canadian, hereditary cancer syndrome, breast cancer, ovarian cancer, cancer predisposing gene

## Abstract

**Simple Summary:**

The French Canadian population of the province of Quebec has been investigated because of its genetic attributes and is known for making significant contributions to the medical genetics field. Their unique genetic background has been attributed to a small number of early settlers from France that contributed to the majority of the gene pool. The French Canadian population has been investigated for the role of known breast and ovarian cancer predisposing genes, such as *BRCA1* and *BRCA2*. In this review we describe the merits of studying this population with respect to the discovery of new such cancer predisposing gene.

**Abstract:**

The French Canadian population of the province of Quebec has been recognized for its contribution to research in medical genetics, especially in defining the role of heritable pathogenic variants in cancer predisposing genes. Multiple carriers of a limited number of pathogenic variants in *BRCA1* and *BRCA2*, the major risk genes for hereditary breast and/or ovarian cancer syndrome families, have been identified in French Canadians, which is in stark contrast to the array of over 2000 different pathogenic variants reported in each of these genes in other populations. As not all such cancer syndrome families are explained by *BRCA1* and *BRCA2*, newly proposed gene candidates identified in other populations have been investigated for their role in conferring risk in French Canadian cancer families. For example, multiple carriers of distinct variants were identified in *PALB2* and *RAD51D*. The unique genetic architecture of French Canadians has been attributed to shared ancestry due to common ancestors of early settlers of this population with origins mainly from France. In this review, we discuss the merits of genetically characterizing cancer predisposing genes in French Canadians of Quebec. We focused on genes that have been implicated in hereditary breast and/or ovarian cancer syndrome families as they have been the most thoroughly characterized cancer syndromes in this population. We describe how genetic analyses of French Canadians have facilitated: (i) the classification of variants in *BRCA1* and *BRCA2*; (ii) the identification and classification of variants in newly proposed breast and/or ovarian cancer predisposing genes; and (iii) the identification of a new breast cancer predisposing gene candidate, *RECQL*. The genetic architecture of French Canadians provides a unique opportunity to evaluate new candidate cancer predisposing genes regardless of the population in which they were identified.

## 1. Introduction

Over the past 40 years, genetic epidemiology studies of breast cancer (BC) and ovarian cancer (OC) have provided unequivocal evidence for the role of genetic factors conferring risk for these diseases. Indeed, the estimated heritability of BC at 31% (95% confidence interval (CI) = 11–51%) and OC at 39% (95%CI = 23–55%) are among the highest for all cancer types [1]. That heritable risk factors are involved is reflected in the familial aggregation of these diseases, where females with at least one first-degree relative with BC have a two-fold increased lifetime risk of BC (relative risk = 1.7; 95%CI = 1.4–2) and those with one first-degree relative with OC have a four-fold increased risk for OC (relative risk = 4.6; 95%CI = 2.1–8.7) [2]. Evidence for the role of specific genetic factors conferring risk for these cancers culminated with the discoveries of *BRCA1* [3] and *BRCA2* [4], the BC and OC cancer predisposing genes. They were identified using a genetic linkage analysis and positional cloning approach that took advantage of multigenerational cancer families featuring premenopausal BC cases (hereditary BC (HBC) syndrome) with or without at least one OC case (hereditary BC and OC (HBOC) syndrome) and having a family structure consistent with the transmission of an autosomal dominant trait (reviewed in [5,6] (Appendix B). *BRCA1* and *BRCA2* have been established as high risk cancer predisposing genes, as heterozygous carriers of pathogenic variants (PVs; all variants in this review are germline unless otherwise stated) have absolute risks greater than 60% for BC and 13–58% for OC, depending on the gene involved [7]. Thus, carriers of a PV have a significantly higher risk for cancer as compared to the overall lifetime risk of BC at 12.9% and OC at 1.3% for North Americans [8]. The spectrum of PVs is multifaceted, where the genetic alteration could affect any region of *BRCA1* or *BRCA2*, and over 2000 different PVs have been identified in each gene in different populations worldwide [9]. *BRCA1* and *BRCA2* are considered major cancer predisposing genes as they account for a significant proportion of HBC and HBOC syndrome families in all studied populations [10,11]. However, not long after their discovery in the mid 1990s [3,4], it became apparent that not all such cancer syndrome families could be explained by *BRCA1* and *BRCA2*, suggesting that other high-risk genes have yet to be discovered [12,13,14].

The availability of a limited number of large multi-generational cancer families and thus the small chance of meiotic recombination events to help refine chromosomal regions for identifying gene candidates posed considerable challenges for discovering *BRCA1* and *BRCA2*. With the completion of The Human Genome Project [15] and advances in understanding the biology of these genes, other strategies, largely favouring a candidate gene approach, have been applied in the identification of new candidate hereditary factors. With each new gene candidate reportedly accounting for only a small proportion of the remaining unexplained cancer syndrome families, it is apparent that another major BC and/or OC predisposing gene like *BRCA1* and *BRCA2* is unlikely. The rarity of carriers of PVs in new risk genes and the genetic heterogeneity of HBC and HBOC syndromes likely explain the difficulty of both identifying and establishing the role of new cancer predisposing gene candidates.

Gene discovery could be facilitated by investigating genetically unique populations that exhibit founder effects due to shared ancestry. A founder effect occurs when a small group of individuals have become isolated from the general population but continue to expand, resulting in a loss of genetic diversity due to genetic drift [16,17]. By chance, genetic drift can result in a significant increase in the frequency of carriers of specific rare disease-associated variants in populations [18]. In the context of HBC and/or HBOC predisposing genes, founder effects have been documented in the Ashkenazi Jewish of Eastern European ancestry [19], Icelandic [20], Finnish [21], and French Canadian (FC) of the province of Quebec, Canada [22,23] populations. In contrast to the general population, all of these populations have been shown to exhibit a limited spectrum of PVs in *BRCA1* and *BRCA2* [24]. Populations exhibiting founder effects have also provided an efficient and cost-effective means to investigate gene candidates in large pools of cancer cases and controls as carriers are readily identifiable due to targeted analyses of PVs [25,26]. However, FCs exhibit a broader array of PVs in *BRCA1* and *BRCA2*, each associated with different carrier frequencies, in contrast to three PVs in *BRCA1* or *BRCA2* in the Ashkenazi Jewish population [27,28] and one PV in *BRCA2* in the Icelandic population [29] (reviewed in [24]).

We posit that the unique genetic architecture of FCs of Quebec provides an opportunity to evaluate new candidate BC and OC predisposing genes (Appendix C). To elaborate upon this working hypothesis, we reviewed studies of FCs of Quebec that described rare variants (minor allele frequency ≤1% in the general population) in known and new candidate cancer predisposing genes that had been identified in the context of HBC and/or HBOC syndrome families consistent with an autosomal dominant mode of inheritance. We also include new interpretations of missense and splice site variants predicted by selected high performing computational tools [30] (Appendix A). We examined the merits of investigating this genetically unique population, especially for characterizing new cancer predisposing gene candidates. We begin this review by summarizing the methods that have been successfully used to identify new BC and OC predisposing gene candidates.

## 2. Methods Applied in the Identification of HBC and/or HBOC Syndrome Predisposing Gene Candidates

Depending on the population studied, between 5% and 40% of HBC and HBOC cancer syndromes families have not been accounted for by PVs in *BRCA1* and *BRCA2* [5,31,32,33,34]. Although the wide range in proportion of *BRCA*-negative families has been attributed to different criteria used to define cancer families, a consistent feature among these reports is that HBC syndrome families are more likely *BRCA*-negative than HBOC syndrome families (Figure 1a,b) [12,35]. Indeed, the research community initially debated the significance of pursuing new high-risk genes in *BRCA*-negative HBOC families, as these families could be due to chance clustering of cancer cases [12,36]. The search for “*BRCA3*” began in earnest in the mid-1990s but the paucity of promising leads suggested that another major cancer predisposing gene explaining the remaining *BRCA*-negative cancer families was unlikely. Although linkage analyses identified promising chromosomal regions, they were unique to the population in which they were identified [37,38]. These observations suggested that HBC and HBOC syndrome families were more genetically heterogenous than previously expected, suggesting that the carrier frequencies of each high-risk gene candidate would be considerably lower relative to *BRCA*-carriers. This working hypothesis led to a variety of gene discovery studies which differed based on case selection, methodology, and analyses.

Observing a higher frequency of carriers of variants in familial cancer cases versus either unselected cancer cases or cancer-free controls is often the first step in proposing new gene candidates. In addition to the availability of participants, identifying *BRCA*-negative families suitable for gene discovery remains an obstacle due to their rarity. It has been estimated that the proportion of families with at least two first-degree relatives with BC or OC is approximately 8% and 2%, respectively, in the general population, regardless of *BRCA1* or *BRCA2* carrier status [43]. Over the past 20 years, national and international consortia have been developed to increase the pool of both familial and sporadic cancer cases and cancer-free controls suitable for research, a concept that was in part successful in identifying cases suitable for *BRCA1* discovery and subsequent validation studies [44,45]. Some examples include The German Consortium for Hereditary Breast and Ovarian Cancer (GC-HBOC), which was established in 1996 (health-atlas.de/projects/2), The Breast Cancer Association Consortium (BCAC) (bcac.ccge.medschl.cam.ac.uk) and The Ovarian Cancer Association Consortium (OCAC), which were established in 2005 (ocac.ccge.medschl.cam.ac.uk), The Japanese HBOC Consortium (JHC) and The Asian BRCA Consortium [46], which were established in 2012 [47], and The Latin American Consortium for HBOC (LACAM), which was established in 2019 [48].

Since the discovery of *BRCA1* and *BRCA2*, 12 new cancer predisposing genes have been proposed to play a role in *BRCA*-negative HBC/HBOC cancer syndrome families (Appendix A). These genes were identified using a candidate gene approach based on the knowledge that BRCA1 and BRCA2 proteins function in the repair of double stranded DNA breaks by homologous recombination (HR) (reviewed in [5,31,34,49]). As examples, *ATM* [50], *BARD1* [31,51,52], *BRIP1* [53], *CHEK2* (whereby a genetic linkage analysis was used in combination with a candidate gene approach) [45], *PALB2* [54], *RAD51C* [55], and *RAD51D* [56] were selected as plausible candidates because they either directly interact with BRCA1 or BRCA2 or are involved at some level in the HR DNA repair pathway [34]. Most of these candidates were identified by investigating *BRCA*-negative families with at least three BC cases in HBC syndrome families, as with *ATM* [50], *PALB2* [54], *CHEK2* [45], *BRIP1* [53], and *RECQL* [57], or families with at least two BC cases and one OC case in HBOC syndrome families, as with *RAD51C* [55] and *RAD51D* [56], attesting to the continuing importance of family-based studies for cancer predisposing gene discovery. A number of these studies, especially in those that characterized candidates in large case–control cohorts, have been facilitated with access to targeted next generation sequencing technologies using gene panels [58]. An important criterion for maintaining candidacy is demonstrating a role for proposed new candidates in independently ascertained cancer cases from the populations in which they were identified and in other populations, as this would strengthen their association with risk of HBC or HBOC [7].

## 3. Genetic Analyses of FC Cancer Cases Facilitate the Interpretation of Variants in *BRCA1* and *BRCA2*

With the identification of *BRCA1* and *BRCA2*, their roles in conferring risk for BC and OC in various populations were investigated by targeted gene sequencing analyses of cancer cases. A complex array of rare variants affecting any coding exon were reported, often unique to the family in which they were identified, initially hindering their clinical interpretation for genetic counselling purposes. The interpretation of variants was facilitated by data sharing where independently identified variants were deposited into databases. The Breast Information Core (BIC) database was the first (no longer actively curated) publicly accessible database for *BRCA1* and *BRCA2* variants identified in cases [59]. The BIC database also included information concerning the ethnic or geographic origins of the variant carriers found useful for medical geneticists. As more variants were deposited in the BIC database and new computational tools became available to predict biological effects, it was possible to infer their clinical relevance for carriers. The BIC database has since been supplanted by ClinVar (ncbi.nlm.nih.gov/clinvar/) [60] and BRCAExchange (brcaexchange.org) [9]. ClinVar also provides inferences of the clinical relevance for a variety of cancer predisposing genes and other risk genes, though information about ancestry is not usually included [60].

As described below, it is clear that *BRCA* variants found to occur in FCs were also reported in other populations, particularly from those with Western European ancestry. In the next section, we describe the unique spectrum of PVs identified in FC BC and OC families and cases and evidence to support that those carriers of the same variant could be due to common ancestors in the FC population, and relate these observations to other studied populations.

### 3.1. Haplotype Analyses Suggest Common Ancestors of Frequently Occurring BRCA1 and BRCA2 Variants in the FC Population

Genetic studies of populations with unique genetic architecture, such as the FCs, have provided important insights into evolutionary origins of PVs in *BRCA1* and *BRCA2*, which have also impacted medical genetic testing practices of these genes. In 1994, the first report of PVs in *BRCA1* in FCs described multiple carriers of BRCA1 c.4327C > T; p.Arg1443Ter (historically known as C4446T), which was attributed to the possibility of shared common ancestors in this population [61]. In 1995, a report of an unusually large FC family with 21 cases of BC showed linkage to the *BRCA2* locus on chromosome region 13q12 [62]. Following these initial reports, and with the discovery of *BRCA2* [4], genetic analyses of *BRCA1* and *BRCA2* in larger defined cohorts of FC HBC and HBOC families identified a limited number of PVs in the FC population [63]. Haplotype analysis of FCs harbouring the most commonly occurring PVs [63] suggested that carriers of each specific variant likely shared a common ancestor.

Haplotyping is a form of genotyping analysis that makes use of polymorphic genetic markers (usually single nucleotide polymorphisms) to investigate genetic regions harbouring rare potentially PVs. The similarity of a variant-bearing haplotype in carriers might indicate identity by descent, having inherited sequences from a common ancestor, in contrast to carriers who share similar nucleotide sequences and are identical by state. Moreover, the size of a haplotype can aid in determining the age of a rare variant in a population. In FCs, the average size of chromosomal regions suggesting identity by descent is 21.3 centimorgans as compared to 8 centimorgans in individuals of North-Western European origin [64], which is not surprising given that many of the present day FCs can be genealogically traced back to common ancestors [65] (Appendix C). There is no evidence to suggest that variants have arisen independently in FCs as different haplotypes of PV-bearing alleles have not been identified.

A more likely hypothesis is that frequently occurring PVs in *BRCA1* and *BRCA2* are the consequence of common ancestry. *BRCA1* (NM_007294.4): c.4327C > T; p.Arg1443Ter remains the most common variant reported in the FC population (Table 1, Figure 2), and haplotype analysis suggests a common ancestor in this population for carriers of this variant [66]. Genealogical reconstruction suggested that carriers of this *BRCA1* variant could be traced to a couple from France and Portugal that were married in 1761 in Quebec. Interestingly, this variant is also one of the most common PVs in *BRCA1* reported in North American populations of Western European ancestry [39]. Haplotype analysis of FC and other populations suggests that *BRCA1* c.4327C > T; p.Arg1443Ter may have arisen independently in different populations [66].

Similar observations have been made for the most commonly reported variants in *BRCA2* (NM_000059.4), c.5857G > T; p.Glu1953Ter (historically known as G6085T) and c.8537_8538del; p.Glu2846GlyfsTer22 (historically known as 8765delAG) (Table 1, Figure 2). Haplotype analysis also suggests that carriers of these variants likely shared a common ancestor in the FC population [63]. Furthermore, haplotype analysis showed that ancestral origins of FC carriers of 8765delAG likely differed from carriers of the same variant reported in the Yemenite Jewish and Sardinian populations, which likely have arisen independently of each other [67,68]. These observations are not surprising due to the purported increased mutability of the AG dinucleotide repeat sequence of exon 20 of *BRCA2* where this variant resides [68].

Other specific PVs in *BRCA1* and *BRCA2* have also been reported in unrelated FCs but occur less frequently in FC cancer families than those described above (Appendix C, Table 1, Appendix A). Among these PVs, haplotype analysis has suggested that carriers of *BRCA2* c.3170_3174del; p.Lys1057ThrfsTer8 (historical name 3398del5), as an example, also shared a common FC ancestor [69].

While loss-of-function variants are readily interpretable for their potential to affect risk, missense variants are more difficult to understand. The genetic architecture of the FC population has been useful in classifying such variants. For example, the rare missense *BRCA2* c.9004G > A; p.Glu3002Lys reported in a number of unrelated cancer families from the North American population was initially classified as a variant of uncertain clinical significance in the BIC database [70]. In addition to identifying this variant in unrelated FC cancer cases, it was shown to segregate with cancer cases in FC HBC families, suggesting that it might indeed be pathogenic [71]. This interpretation was supported by subsequent in cellulo assays revealing that it encoded a protein with aberrant HR function, and this finding led to its reclassification as pathogenic [72].

Identifying frequently occurring variants in populations that have undergone genetic drift, such as the FCs, is important as it supports the notion that PVs in *BRCA1* and *BRCA2* are least likely to arise from de novo mutagenesis in the germline. Indeed, there are no credible reports of de novo germline variants in these genes, though there is evidence that PVs can arise due to this mechanism for other high-risk cancer predisposing genes, such as those reported in *RB1* in the rare non-hereditary forms of pediatric retinoblastoma [73]. The stable origin of heritable PVs also provides a means of cost-effective genetic testing for PVs found most commonly in founder populations for research and medical genetic purposes, the exemplars being the three PVs in *BRCA1* and *BRCA2* found to account for almost all *BRCA*-carriers in the Ashkenazi Jewish population [74]. As shown in an early targeted analysis of 20 variants in FCs, 84% of *BRCA1* and *BRCA2* positive HBC and HBOC syndrome families harbour one of five specific PVs in these genes accounting for the high frequency of these PVs observed in BC- and OC-affected individuals in this population [35]. Unlike the high (1.1–2.5%) carrier frequency of the three founder *BRCA1* and *BRCA2* PVs observed in the Ashkenazi Jewish population [75,76,77], there is no evidence to suggest that the overall *BRCA*-carrier frequency in FCs of Quebec is higher than 0.25% carrier frequency estimated for Northern Americans [78,79]. Indeed, a recent study has shown that *BRCA1* and *BRCA2* variants are rare (<0.2%) in the non-cancer FC population with no personal or family history of cancer relative to cancer cases [80].

### 3.2. The Spectrum of BRCA1 and BRCA2 Variants in FCs

Genetic studies of the FC population have helped validate the role of rare potentially PVs in *BRCA1* and *BRCA2* in cancer syndrome families, sporadic cancer cases regardless of family history of cancer, and the general population. The overall frequency of *BRCA*-carriers in FCs with sporadic BC [81,82] or OC [83] (Figure 1c) is within the range reported for BC (5–10%) and OC (12–15%) from North American, European, and other populations [51,84]. Since the initial reports of *BRCA1* and *BRCA2* PVs in FCs, the spectrum of frequently occurring variants identified in FC BC and/or OC cases has expanded to a total of 25 variants, including 18 PVs [35,39,63,80,81,82,83,85,86,87,88,89,90,91] (Table 1, Figure 3, Appendix A), of which the majority are nonsense and frameshift variants that are expected to result in the loss of the protein function.

In reviewing the literature, 36 rare variants have been reported only once in *BRCA1* and *BRCA2* in FCs with BC or OC [35,63,87,89,99] (Appendix A). The ease of gene sequencing enabled the identification of new variants in the FC population using targeted gene sequencing of all exons and splice site regions. Of these 36 variants, the most promising PVs are the 11 that are classified as pathogenic and 13 of uncertain significance, the remainder being benign based on ClinVar [60] and American College of Genetics and Genomics (ACMG) guidelines (Figure 3, Table 1, Appendix A). These variants include 11 missense, 6 frameshift, 4 nonsense, 2 splicing, and 1 in-frame deletion. The classification of the majority of these variants is consistent between in silico tools and functional characterizations, though some are not, which is in line with the approximately 90% accuracy of these tools [30] (Appendix A). Although none of the in silico splicing tools predicted that *BRCA1* c.81-6T > C affects splicing, a biological assay has shown that there is an effect on RNA splicing [100] (Appendix A). This is not surprising as the in silico tools used in that study, which predicted that this variant would affect splicing, differed from those applied in this review. Currently, the splicing tools that have the best predictive performance have not been systematically investigated, unlike the established list of best performing in silico tools suggested for classifying missense variants [30]. *BRCA2* c.7007G > A; p.Arg2336His is predicted to affect splicing by all four splicing tools used (Appendix A). Of the 11 missense variants, seven were potentially pathogenic using our in silico tools (Appendix A). Of note, two of these missense variants, *BRCA1* c.736T > G; p.Leu246Val and *BRCA2* c.8850G > T; p.Lys2950Asn, did not affect the function of the HR pathway [101,102] (Appendix A). Although *BRCA2* c.9976A > T; p.Lys3326Ter introduces a stop codon predicted to truncate the BRCA2 protein, its clinical significance remains controversial (reviewed in [103,104]), and this is due to: (1) the fact that its carrier frequency at 0.6% in the general population, though rare, is higher than that of other PVs in *BRCA2* (0–0.001%) (Appendix A); and (2) its location in the C-terminus where it has been proposed to exert the least effect on the function of the protein [105]. Independent studies of sporadic and familial cancers have shown an increased risk for BC and OC in carriers of this *BRCA2* variant [105,106]. Although this variant has not been investigated to the same extent as other PVs in the FC population, targeted gene sequencing analysis identified *BRCA2* c.9976A > T in two out of 256 (0.8%) unrelated HBC syndrome families [87], placing it among the least frequently occurring *BRCA2* PVs in FC cancer families.

Complex and difficult to detect large deletions or genomic rearrangements in *BRCA1* and *BRCA2*, which are rarely found in the general population, are also likely rare in FCs, as suggested by a study of BC and OC cases from cancer families that applied the established multiplex ligation probe amplification (MLPA) analysis technique and found no examples of carriers of such variants [107]. As observed with variants in other populations, there is no obvious clustering of PVs in any protein encoding or splice site region of *BRCA1* or *BRCA2* (Figure 3). PVs that occur in the defined BC Cluster and OC Cluster Regions in *BRCA1* and *BRCA2* have been statistically associated with increased risk of BC, or OC, respectively [108]. However, this has not been studied in FCs due to the overall low frequency of carriers in this population enabling statistical associations of each variant with risk of BC or OC (Figure 3).

The spectrum of *BRCA1* and *BRCA2* PVs described in the FC population is not surprising given the European origins of FC ancestors. Indeed, all PVs identified in FCs in *BRCA1* and *BRCA2* have been reported in other populations (Appendix A). Early studies from our group showed that carriers of the most commonly reported *BRCA1* c.4327C > T; p.Arg1443Ter had grandparents with ancestral ties to different geographic regions across Quebec [63], whereas carriers of the less frequently reported variants were each from a smaller defined region within Quebec even though they were identified in unrelated families [63,69,71].

The unique genetic architecture of FCs thus affords an opportunity to investigate this population for new candidate variants in known cancer predisposing genes as well as help validate new cancer predisposing genes for heritable BC and OC, as elaborated upon further below.

## 4. Genetic Analyses of FC Cancer Cases Helps Define the Role of New Candidate HBC/HBOC Predisposing Genes

The genetic analyses of new cancer predisposing genes in the FC population has provided support for their role in hereditary BC and OC (Table 2, Appendix A). In this section, we describe studies of FCs involving PVs in new risk genes, especially those associated with HBC and HBOC syndrome families exhibiting an autosomal dominant mode of inheritance [34]. However, before doing so, it is important to mention the few studies of FC HBC and HBOC families involving established cancer predisposing genes, *TP53* and *STK11*, which are known to play a role in Li–Fraumeni (MIM:151623) [109] and Peutz–Jeghers (MIM:175200) [110,111] syndrome families, respectively. These genes are plausible candidates to account for the fact that *BRCA*-negative HBC families as heterozygous carriers of PVs in these genes also have significant absolute risks for premenopausal BC: exceeding 60% for *TP53* carriers [7] and 40–60% for *STK11* carriers [7]. Our group has reported seven rare variants in *TP53* in FC familial or sporadic BC cases [41,112], where five are classified as PV by *in silico* analyses. Interestingly, there were two carriers of the same PV in *TP53* (NM_000546.6), namely c.638G > A; p.Arg213Gln or c.685T > C; p.Cys229Arg, among the BC cases, where the latter PV was identified in cases not known to be related to each other [41,112]. The overall estimated carrier frequency of PVs in *TP53* in HBC families at 3.8% [41] (Figure 1f) and 1.2% in sporadic BC cases [112] was higher than expected given the estimated 1 in 5000 to 20,000 TP53-carriers in the general population worldwide (reviewed in [113]). The carrier frequency of PVs in *TP53* in the FC population and whether the carriers of the same PV share a common ancestor have yet to be determined. Only one rare variant in *STK11* was identified in a study of 96 *BRCA*-negative HBC families, where the carrier family did not exhibit clinical features consistent with Peutz–Jeghers syndrome [114] (Appendix A). This *STK11* missense variant (c.1062C > G; p.Phe354Leu) is benign/likely benign in ClinVar and was not predicted to be a PV based on our in silico analysis. There are no reliable estimates of the carrier frequency of *STK11* PVs in HBC or HBOC families, though it is likely rarer than for carriers of *TP53* variants. While there are other studies of established cancer predisposing genes in FC cancer cases, such as those involved in Cowden syndrome (MIM:158350) and Lynch syndrome (MIM:120435; 609310), which also feature BC and OC cancer, they have not been systematically explored in *BRCA*-negative HBC and HBOC families or in BC or OC cases, and thus are not included in this review. Regardless, it is apparent from studies of FCs and of other populations that established BC and OC cancer predisposing genes are implicated in a small proportion (0–1%) of *BRCA*-negative cancer families, further supporting the hypothesis that other cancer predisposing genes have yet to be discovered.

### 4.1. A Predominant PV in PALB2 Frequently Occurs in FC Hereditary BC Cases

*PALB2* is the most promising of the newly proposed BC predisposing genes [54]. In 2007, after an independent report described PALB2 as a BRCA2 binding partner [115], targeted sequencing of *PALB2* as a new candidate gene for hereditary BC in Finnish HBC families determined a statistical association with one of the identified loss-of-function variants in this gene [54]. Further targeted genotyping analysis showed that the carrier frequency of this variant was higher in BC cases versus controls in the Finnish population (Appendix A). Soon thereafter, targeted sequencing analysis of *PALB2* in 50 FC early-onset or familial BC cases identified one carrier of *PALB2* (NM_001005735.2): c.2323C > T; p.Gln775Ter, and this variant was also identified in 2/356 BC cases but not in controls [116]. Carriers were subsequently identified in 2% of FC HBC families *BRCA*-negative for the five most commonly occurring PVs observed in FCs (Table 2, Figure 1d and Figure 2). This variant is predicted to introduce a stop codon at amino acid position 775 of *PALB2*, and if expressed would render the truncated protein non-functional, suggesting its role in pathogenicity [116]. Haplotype analysis of unrelated FC carriers suggested that they may have inherited this *PALB2* variant from a common ancestor [40,88,114]. *PALB2* c.2323C > T accounts for 0.7% of FCs with early-onset BC not selected for family history of BC [88]. In contrast, carriers of this *PALB2* variant are rare in cancer-free FC controls, as none were found in approximately 2000 cancer-free individuals [80,88]. *PALB2* c.2323C > T is the first example of a newly proposed BC risk gene shown to play a role in *BRCA*-negative HBC syndrome families in the FC population.

While there is mounting evidence from the research community supporting *PALB2* conferring an increased risk for BC, its role in hereditary OC is unclear [7,117,118]. Our analysis of *PALB2* c.2323C > T in sporadic FC OC cases only identified one carrier who had OC at the age of 58 years among 238 (0.2%) cases (Figure 1e). Interestingly, this carrier also had BC at the age of 52 years [118]. A report of 524 *PALB2* PV carrier families of European ancestry estimated the associated relative risks with BC as 7.2 (95%CI = 5.8–8.8; *p* = 6.5 × 10^−76^) and OC as 2.9 (95%CI = 1.4–6; *p* = 4.1 × 10^−3^) [118]. A targeted sequencing analysis of 54 candidate genes selected based on their function in HR repair in OC and controls by OCAC only identified a statistical association of potentially PVs in *PALB2* with OC [119]. The National Comprehensive Cancer Network (NCCN) guidelines reported estimates that the absolute risk for OC in carriers of *PALB2* PVs is between 3 and 5% compared to the absolute risk for BC: 41–60% [7].

Other *PALB2* variants in FC BC cases have been reported [114,116] that were predicted as potentially pathogenic by our selected in silico tools, with the exception of two variants (Table 2, Appendix A). Large germline deletions or insertions have been investigated in BC cases from FC HBC or HBOC families using MLPA, and though none were identified [114,116], they have been reported in studies of other populations [120,121]. Potentially PVs in *PALB2* have since been reported in diverse populations, providing support for its role in BC risk. A large multi-center study involving familial and sporadic cases from diverse populations estimated that the risk for female BC in *PALB2* carriers to age 80 is 53% (95%CI = 44–63%), adding *PALB2* to the list of validated high-risk BC predisposing genes [118].

### 4.2. A Frequently Occurring Missense Variant in FCs Supports a Role for RAD51D in Hereditary OC

Genes encoding members of the RAD51 family are directly involved in the HR repair pathway and as such have been investigated as plausible new BC and OC predisposing gene candidates in *BRCA*-negative families [34]. The earliest reports appeared in 2010 for *RAD51C* [55] and 2011 for *RAD51D* [56]. The sequencing of protein encoding and splice regions of these genes in *BRCA*-negative HBC and HBOC families identified rare potentially PVs. A higher frequency of carriers from HBOC families with either OC or BC, but not BC cases from HBC families, were found to harbour one of these variants as compared to the controls. Subsequent studies further supported the strong association of PVs in *RAD51C* and *RAD51D* with OC [117,122,123]. Estimates of the absolute risk for OC in carriers of *RAD51C* or *RAD51D* PVs are greater than 10% [7], which is in line with established cancer predisposing genes conferring a significant risk of cancer, as for carriers of PVs in *BRCA1* and *BRCA2*. The relative risk for OC is estimated to be as high as 40% if carriers of variants in these genes have a first-degree relative with OC [117]. However, the role of *RAD51C* and *RAD51D* in BC risk is less clear [7], though population-based studies suggest that carriers of PVs in these genes are more likely found among BCs classified as estrogen receptor-negative or triple-negative [51,52] (tumours defined by the absence of estrogen and progesterone receptor expression accompanied with no overexpression of human epidermal growth factor receptor 2).

Studies of a rare *RAD51D* variant found in FCs provided further evidence in support of this gene playing a role in OC risk. *RAD51D* (NM_002878.4): c.620C > T; p.Ser207Leu was classified at the time of the study as a variant of uncertain significance in ClinVar [60] as it was rare in the general population [122,124,125] (Appendix A). Following initial reports of carriers of *RAD51D* c.620C > T, a number of OC and BC cases from FC HBOC and HBC families were found to carry the same variant in medical genetic units by multi-gene panel testing [42], although an early study of HBC and HBOC families, which included a small number of FC families, did not identify any potentially PVs in *RAD51D* [126] Appendix A). This missense variant was investigated in sporadic BC and OC cases not selected for age of diagnosis to further determine its role in conferring risk for cancer in FCs [42]. The results revealed a significantly higher carrier frequency in OC cases relative to controls (3.8% vs. 0.2%) (Figure 1g and Figure 2, Table 2). In sporadic OC cases, the carrier frequency of this variant was comparable to carriers of *BRCA1* c.4327C > T; p.Arg1443Ter (3.4%), the most prevalent PV in FCs [42]. Interestingly, in this study, there were no co-occurring carriers of these specific *RAD51D* and *BRCA1* variants, nor with the other five most common *BRCA1* and *BRCA2* variants found in FCs [39,42]. In cellulo assays showed that this variant encodes an aberrant protein, RAD51D p.Ser207Leu, that affects the HR pathway function [42], and thus may be pathogenic and play a role in conferring risk for OC in carriers. This is reflected in the conflicting interpretations of *RAD51D* c.620C > T in more recent updates of ClinVar [60] (Accession ID: VCV000142102.11) as a variant of uncertain significance (three submissions), likely pathogenic (five submissions), and pathogenic (two submissions), and a variant of uncertain significance by ACMG guidelines (Appendix A). Haplotype analysis has suggested that carriers of *RAD51D* c.620C > T likely shared a common ancestor [42]. The high frequency of carriers of *RAD51D* c.620C > T in the OC cases was not expected given the rarity of carriers of this missense variant or any PV in this gene in FC and other non-FC populations [124]. These findings place *RAD51D* c.620C > T; p.Ser207Leu among one of the most commonly observed PVs conferring risk of OC in FCs.

Less is known about *RAD51C* in FCs, though one report described a targeted sequencing analysis of this gene in 152 *BRCA*-negative FC HBC and HBOC families [127]. In this report, no loss-of-function or potentially pathogenic missense variants were identified, suggesting that these variants in *RAD51C* are rare in this population and may not significantly contribute to FC HBC and HBOC families.

### 4.3. BRIP1 and CHEK2 in FC BC and OC Cases

Relative to *BRCA1* and *BRCA2*, less is known of the role of *BRIP1* and *CHEK2* cancer predisposing genes in conferring risk of BC and OC in FCs. Studies have shown that variants in these genes likely confer risk of BC that is lower than for PVs in *BRCA1*, *BRCA2*, *PALB2*, *RAD51C*, and *RAD51D* [45,53,128]. These genes were plausible BC and/or OC predisposing candidates because of their role in directly interacting with BRCA1 protein through its binding to BRIP1 protein [129] or playing a central role in the cellular response to double stranded DNA breaks, as shown with CHEK2 protein (reviewed in [34,130,131]). Targeted sequencing analysis of *BRIP1* in FC BC cases with a family history of BC identified missense variants [132], where three of them are predicted as potentially PVs by our in silico analysis (Appendix A, Appendix A). A population-based study by OCAC investigating sporadic cancer cases estimated that carriers of *BRIP1* variants had a relative risk of 11.2 for OC (95%CI = 3.2–34.1; *p* = 1 × 10^−4^) [133], but not for BC [51,52]. The NCCN guidelines reported an estimated absolute risk for OC in carriers to be greater than 10%, but had limited evidence-based data on the risk for BC [7]. In one study of FCs, the frequency of *CHEK2* (NM_007194.4): c.1100del; p.Thr367MetfsTer15, was 2% in BC families with at least two or more BC cases diagnosed before the age of 65 years, which is lower but comparable to the 3.7% carrier frequency reported in BC families from the general population [45] (*p* = 0.7 using Fisher’s exact test). In another study, the carrier frequency of this variant was 1.1% in FCs with young age of onset sporadic BC [88], which is comparable to the 1% carrier frequency reported in the general population [134]. Targeted gene sequencing analyses identified other *CHEK2* variants in FC BC cases, with c.1217G > A; p.Arg406His being the most promising candidate based on our in silico analysis (Table 2, Appendix A), though the carrier frequencies were not significantly different between BC cases and controls [135]. The absolute risk for BC in *CHEK2* PV carriers has been estimated to be in the range of 15–40%, but with no evidence for increased risk for OC [7]. The identification of *BRIP1* and *CHEK2* variants in FC HBC or HBOC syndrome families, which have also been reported in BC and OC cases from other populations (Appendix A), supports their role in the risk of these cancers.

### 4.4. The Role of Proposed Cancer Predisposing Genes in FCs

New candidate genes where the association with HBC and/or HBOC is still unknown include *BARD1*, *MRE11*, *RAD50*, and *NBN* (Appendix A). *BARD1*, which encodes a BRCA1 interacting protein, was proposed as a candidate risk gene soon after the discovery of *BRCA1* and *BRCA2* [31,136,137]. *MRE11*, *RAD50*, and *NBN*, which were proposed as candidates a decade after the discovery of *BRCA1* and *BRCA2* [138,139], encode proteins of the MRN complex, a multi-protein structure that has been shown to play an important role in sensing double stranded DNA breaks for DNA repair (reviewed in [34]). Thus far, there have been reports investigating variants in *BARD1*, *MRE11*, and *NBN* in FCs, but not *RAD50*. Four rare variants in *BARD1* in BC cases from FC HBC families have been described that were identified using targeted sequencing analyses [140]. Two *BARD1* missense variants have been classified as likely benign or benign based on ACMG guidelines, which is consistent with the prediction with our in silico analysis (Table 2, Appendix A). *BARD1* (NM_000465.4): c.1075_1095dup; p.Leu359_Pro365dup was found in four unrelated BC cases (4/96 cases vs. 2/87 controls), though this observation was not surprising given the genetic architecture of FCs [140]. The sequencing of *NBN* in BC cases with a family history of BC identified carriers of several different variants where none were found in more than one FC BC case [141]. The one missense variant *NBN* (NM_002485.5): c.283G > A; p.Asp95Asn reported in this study has been classified as a variant of uncertain significance, though it was predicted as potentially pathogenic by our in silico analysis (Appendix A). However, a recent study of *MRE11* c.1516G > T; p.Glu506Ter, which has been reported in multiple FC cancer cases and also found in other populations, suggested that it may not to be associated with BC risk [142], which is in line with recent findings from a large BC case–control study [52] (Table 2, Appendix A). In a study of FCs, this rare loss-of-function variant was not identified in 1920 BC and 341 OC cases but in 4/1891 (0.2%) adult cancer-free controls and 1/1932 (0.01%) newborns. Though the differences in *MRE11*-carrier frequency between cancer cases and controls were not significant, the findings questioned its classification as a PV (Appendix A). Immunohistochemistry analysis of BC and OC tumours from carriers showed strong protein expression of MRE11, and genetic analyses of tumour tissues revealed the presence of both parent of origin *MRE11* alleles [142]. With these findings, the authors of this study questioned the candidacy of *MRE11* as a BC predisposing gene. Thus, the unique genetic architecture of FCs may also aid in the resolution of potentially benign candidate variants found to occur in other populations. With the exception of one *BARD1* variant, all of the rare variants in *BARD1*, *MRE11*, and *NBN* that were identified in FC cancer cases have also been reported in studies of BC and OC from other populations (Appendix A).

## 5. Discovery of New Candidate HBC/HBOC Predisposing Genes Identified in the FC Population

*RECQL* was identified as a new candidate BC predisposing gene by focusing on the study of FC and Polish cancer cases as both of these populations exhibit genetic drift, suggesting a genetic architecture amenable for the investigation of new candidate genes [57]. A complex but effective strategy was used that included the genetic analyses of three independent BC groups per population (Appendix A). With respect to the studies of FC study groups, whole exome sequencing was performed on 51 index BC cases selected from HBC/HBOC syndrome families and/or if they were diagnosed at a young age. All BC cases were not carriers of PVs in *BRCA1*, *BRCA2*, *PALB2*, *CHEK2*, and *NBN* that frequently occur in FC and Polish populations. Rare loss-of-function variants were prioritized as top candidates if they were identified in at least two unrelated BC cases. Using this approach, three carriers of loss-of-function variants in *RECQL* were identified: two in index cases from HBOC families (Table 2, Appendix A). These findings were then validated by targeted sequencing in 475 familial BC cases who were not carriers of any of the most commonly occurring *BRCA1* and *BRCA2* PVs found in FCs. Though no additional carriers of the candidate variants were found, two carriers of a newly identified loss-of-function variant *RECQL* (NM_032941.2): c.643C > T; p.Arg215Ter were identified (Table 2). Targeted genotyping of *RECQL* c.643C > T in a third group of FC BC cases with a family history of this disease or who had BC before the age of 50 years identified additional carriers (5/538; 0.93% in cases vs. 1/7136; 0.014% in controls; *p* < 0.0001 using Fisher’s exact test). This strategy, replicated with the Polish study groups, identified three potentially pathogenic *RECQL* variants, which differed from those found in FCs, and one was found in unrelated cases in this population [57]. The journal that published the identification of *RECQL* as a candidate BC predisposing gene in FC and Polish populations also reported the identification of the same candidate gene by studying the Chinese Han BC population [143]. All potentially pathogenic *RECQL* variants identified in Chinese Han cancer cases differed from those found in the study of the FC and Polish populations. The clinical significance of *RECQL* has yet to be determined, though subsequent genetic studies of *RECQL* have been conflicting, even questioning its role in BC predisposition [144]. This may be a result of not using fully matched control cohorts, which may lead to spurious associations [52]. Noteworthy is that while the focus of research in these *RECQL* discovery studies was focused on identifying new BC predisposing genes, pedigree inspection of *RECQL*-carrier families clearly showed the presence of OC cases [57]. Therefore, further research is necessary to resolve the role of *RECQL* in BC and OC predisposition.

## 6. Perspectives

The contribution of FC participants to the study of a variety of genetic disorders has been recognized by researchers and the health care community [22,23]. Genetic analyses of FCs of Quebec, Canada have contributed to our understanding of the role of rare PVs in cancer predisposing genes that confer high risks for the hereditary form of BC and OC inherited in an autosomal dominant manner. An important consideration in studying populations with unique genetic architecture, such as FCs, is the possibility of identifying carriers of candidate variants that are rare in the general population but can be shown to be benign [145].

More work remains on the role of new BC and OC gene predisposing gene candidates in FCs, such as *FANCC* [146], *FANCM* [147,148], and *RAD50*, that have been described in other populations but have not yet been reported for FCs. Studies of FCs of Quebec have sparked new initiatives geared towards the resolution of rare disease-associated variants identified in this genetically unique population. One of the first population-based cohorts useful for interpreting the frequency of candidate variants included a collection of DNA samples from newborns from the Quebec City area, where the majority of inhabitants are FC [149,150,151], and as described above has been successfully used to study carrier frequencies of PVs in *BRCA1* and *BRCA2* [88], *PALB2* [116], and *RAD51D* [42]. The launch of CARTaGENE (cartagene.qc.ca) [152], which is part of the Canadian Partnership for Tomorrow’s Health (CanPath), a Canada-wide population-based cohort including over 330,000 participants with the objective of improving knowledge about chronic diseases (canpath.ca) [153], has provided population-matched controls for genetic studies. CARTaGENE is a prospective population-based biobank including 43,000 participants from Quebec between the ages of 40 and 69 years, with the aim of improving the prevention, diagnosis, and treatment of chronic diseases, including cancer. The availability of not only biological specimens but also over 600 detailed health, socio-demographic, and physiological metrics allows for the interpretation of genetic data in the context of these metrics. Personal and first-degree family history of cancer is included. Other epidemiological factors, such as oral contraceptive pill use, reproductive factors, and/or hormone replacement therapy, have been associated with BC [154] and OC [32], and these can be investigated in candidate variant carriers. As data become available from these projects, candidate variants identified in cancer families and cases can be investigated in CARTaGENE cohorts to evaluate their frequency in population-matched controls and thereby their relevance as candidates.

## 7. Conclusions

Over the past 25 years since the discovery of *BRCA1* and *BRCA2*, it is increasingly clear that FCs of Quebec, Canada have played a significant role in defining the genetic landscape of cancer predisposing genes. Variants in *BRCA1* and *BRCA2* have been well characterized in FCs and the investigation of other HBC/HBOC predisposing genes has allowed for their identification and characterization. Studies of the FC population have provided evidence that *RECQL* is a new HBC/HBOC predisposing gene. The unique genetic architecture of the FC population should provide the opportunity to identify future cancer predisposing genes.

## Figures and Tables

**Figure 1 cancers-13-03406-f001:**
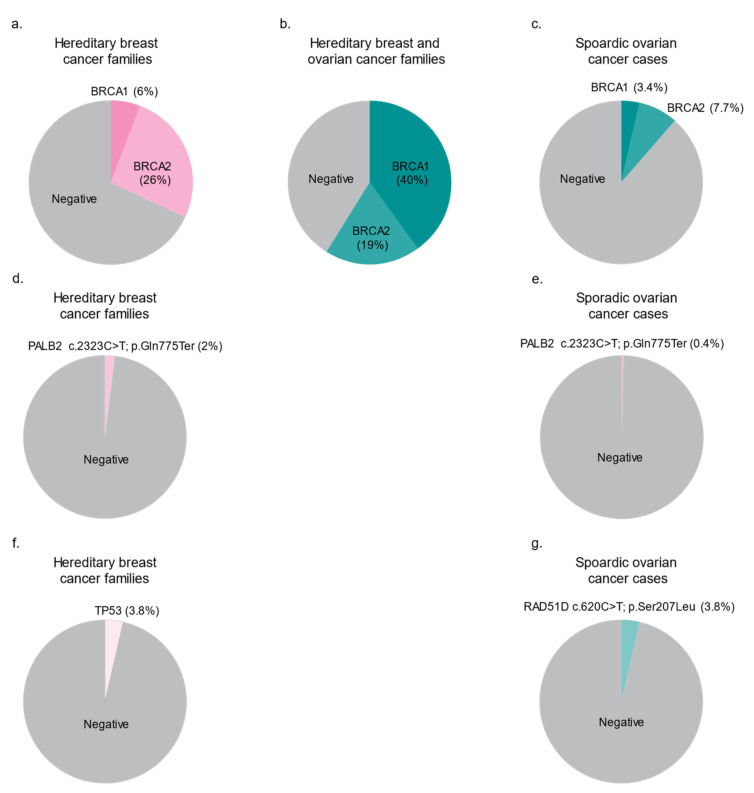
Representative carrier frequencies of frequently occurring pathogenic variants in HBC and HBOC predisposing genes in French Canadians of Quebec. Distribution of *BRCA1* and *BRCA2* variants in hereditary breast cancer syndrome (**a**), hereditary breast and ovarian cancer syndrome (**b**), and sporadic ovarian cancer cases (**c**). Carrier frequency of a *PALB2* variant (**d**) and *TP53* variants (**f**) in hereditary breast cancer. Carrier frequency of a *PALB2* variant (**e**) and a *RAD51D* variant (**g**) in sporadic ovarian cancer cases. Data from [35] where 169 cancer families were analyzed. Selected variants in *BRCA1* (*n* = 11) and *BRCA2* (*n* = 9) were assessed in this study.Data from [39] where 439 sporadic ovarian cancer cases were analyzed. Selected variants in *BRCA1* (*n* = 2) and *BRCA2* (*n* = 4) were assessed in this study. Data from [40] where 48 hereditary breast cancer families and 238 sporadic serous ovarian cancer cases were analyzed. One *PALB2* variant (c.2323C > T; p.Gln775Ter) was assessed in this study. Data from [41] where 52 hereditary breast cancer families were analyzed. Targeted sequencing of *TP53* exons and splice sites was assessed. Data from [42] where 341 sporadic high-grade serous ovarian cancer cases were analyzed. One *RAD51D* variant (c.620C > T; p.Ser207Leu) was assessed in this study. Sporadic ovarian cancer cases are all derived from the same study group [39].

**Figure 2 cancers-13-03406-f002:**
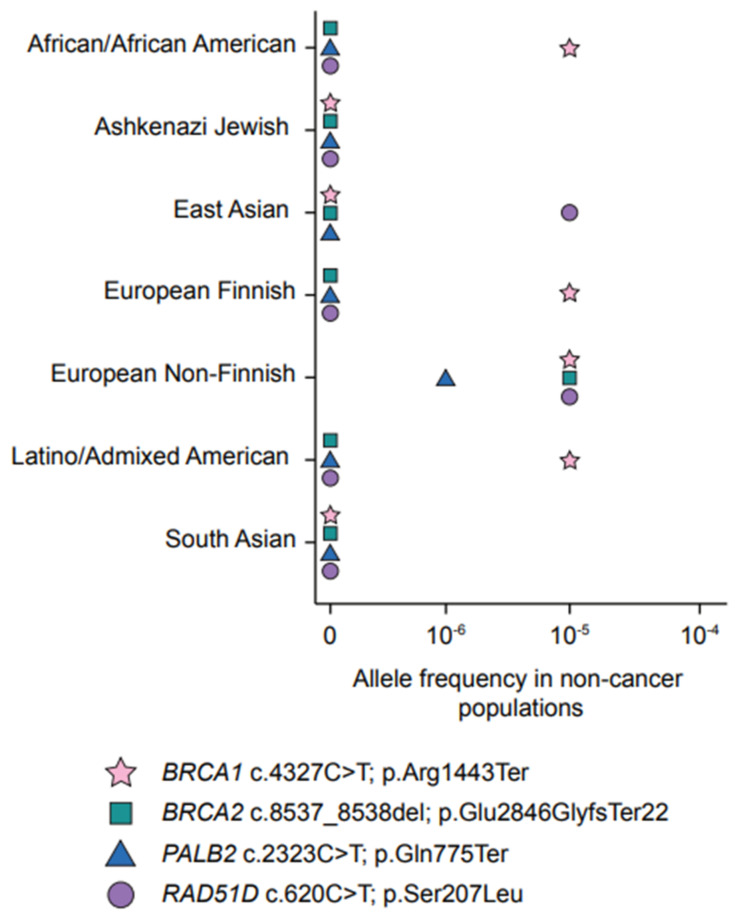
The most frequently occurring pathogenic variants *BRCA1*, *BRCA2*, *PALB2*, and *RAD51D* in French Canadians of Quebec and their allele frequency in other worldwide non-cancer populations. Source of the data: gnomAD v2.1.1 (gnomad.broadinstitute.org).

**Figure 3 cancers-13-03406-f003:**
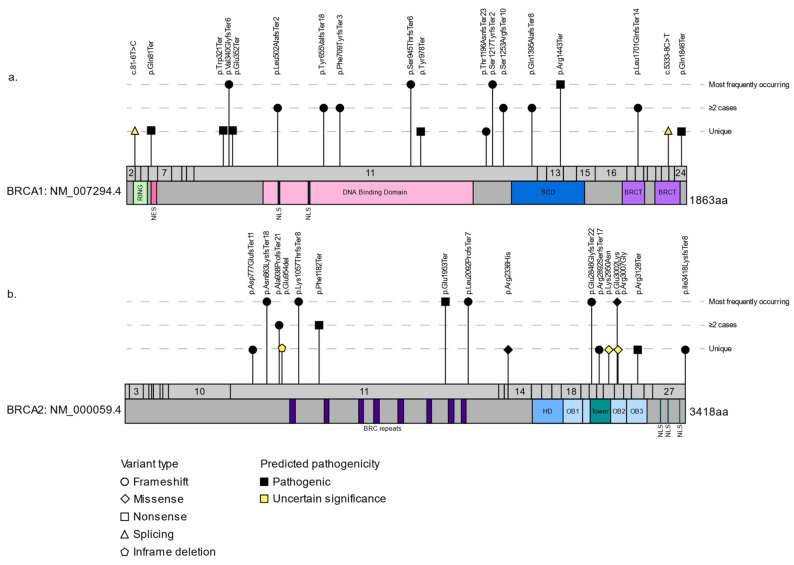
Pathogenic variants and variants of uncertain significance reported in French Canadians of Quebec mapped to full length *BRCA1* (**a**) or *BRCA2* (**b**) transcripts. Variants are predicted to be pathogenic or have uncertain significance based on ClinVar and/or ACMG guidelines. RING = Really Interesting New Gene domain; NES = Nuclear export signal; NLS = Nuclear localization signal (*BRCA1*: [92]; *BRCA2*: [93]); SCD = Serine cluster domain [94]; BRCT = BRCA1 C Terminus domain; BRC repeats = BRCA2 repeats; HD = Helical domain; OB = Oligonucleotide binding; Tower = Domain essential for DNA binding [95]. *BRCA1* GenBank: AAC37594.1 [96], *BRCA2* GenBank: AAB07223.1 [97], DNA binding domain [98]. See Appendix A for more information about variants.

**Table 1 cancers-13-03406-t001:** Frequently occurring pathogenic variants in *BRCA1* and *BRCA2* in French Canadians of Quebec ^1^.

Gene	Coding Change ^2^	Protein Change ^2^	Historical Nomenclature	Shared Haplotype in Carriers	Source(s)
*BRCA1*	c.962G > A	p.Trp321Ter	1081G > A	-	35, 63, 85
	c.1054G > T	p.Glu352Ter	E352X	-	63, 83
	c.1961dup	p.Tyr655ValfsTer18	2080insA	-	83
	c.2125_2126insA	p.Phe709TyrfsTer3	2244insA	-	83, 83, 84, 86
	c.2834_2836delinsC	p.Ser945ThrfsTer6	2953del3 + C	Yes	35, 59, 63,83, 85, 87, 88
	c.3649_3650insA	p.Ser1217TyrfsTer2	3768insA	Yes	35, 59, 86
	c.3756_3759del	p.Ser1253ArgfsTer10	3875delGTCT	-	35, 85, 88
	c.4041_4042del	p.Gly1348AsnfsTer7	4160delAG	-	35, 83
	c.4327C > T	p.Arg1443Ter	C4446T	Yes	35, 59, 63, 77, 79, 77, 83–86, 88
	c.5102_5103del	p.Leu1701GlnfsTer14	5221delTG	-	35, 77, 83
*BRCA2*	c.2588dup	p.Asn863LysfsTer18	2816insA	Yes	35, 59, 83, 86
	c.2808_2811del	p.Ala938ProfsTer21	3034del4	Yes	35, 83, 86
	c.3170_3174del	p.Lys1057ThrfsTer8	3398del5	Yes	35, 84–86, 88
	c.3545_3546del	p.Phe1182Ter	3773delTT	-	35, 63, 84, 86
	c.5857G > T	p.Glu1953Ter	G6085T	Yes	35, 59, 63, 78–79, 84–86, 88
	c.6275_6276del	p.Leu2092ProfsTer7	6503delTT	Yes	35, 59, 78, 83, 86
	c.8537_8538del	p.Glu2846GlyfsTer22	8765delAG	Yes	35, 59, 63, 77–79, 81, 83–86, 88
	c.9004G > A	p.Glu3002Lys	E3002K	-	63, 68, 86

- Data not available. ^1^ See Appendix A for more information on variants. ^2^ All annotated variants are based on the Human Genome Reference GRCh37/hg19 and the Human Genome Variation Society (HGVS) nomenclature guidelines.

**Table 2 cancers-13-03406-t002:** Frequently occurring potentially pathogenic variants in new candidate cancer predisposing genes in French Canadians of Quebec ^1^.

Gene	Canonical Transcript	Coding Change ^2^	Protein Change ^2^	Shared Haplotype in Carriers	Source(s)
*BARD1*	NM_000465.4	c.1075_1095dup	p.Leu359_Pro365dup	-	134
		c.1930G > A	p.Val644Ile	-	134
		c.2212A > G	p.Ile738Val	-	134
*BRIP1*	NM_032043.3	c.577G > A	p.Val193Ile	-	126
		c.2097 + 7G > A	-	-	126
*CHEK2*	NM_007194.4	c.1100del	p.Thr367MetfsTer15	-	85, 105
		c.1217G > A	p.Arg406His	-	129
*MRE11*	NM_005590.4	c.1516G > T	p.Glu506Ter	-	136
*PALB2*	NM_001005735.2	c.226A > G	p.Ile76Val	-	108
		c.1273G > A	p.Val425Met	-	105
		c.1676A > G	p.Gln559Arg	-	105
		c.2323C > T	p.Gln775Ter	Yes	84, 85, 107, 108, 107
		c.2590C > T	p.Pro864Ser	-	105
*RAD51D*	NM_002878.4	c.620C > T	p.Ser207Leu	Yes	119
*RECQL*	NM_032941.2	c.643C > T	p.Arg215Ter	-	53
*TP53*	NM_000546.5	c.638G > A	p.Arg213Gln	-	102, 103
	NM_000546.5	c.703A > G	p.Asn235Asp	-	103
	NM_000546.5	c.730G > A	p.Gly244Ser	-	103
	NM_000546.5	c.742C > T	p.Arg248Trp	-	103
	NM_000546.5	c.844C > T	p.Arg282Trp	-	103

-: Data not available. ^1^ See Appendix A for more information on variants. ^2^ All annotated variants are based on the Human Genome Reference GRCh37/hg19 and the HGVS nomenclature guidelines.

## Data Availability

Not applicable.

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
