# Peer review of "The Genetic Analyses of French Canadians of Quebec Facilitate the Characterization of New Cancer Predisposing Genes Implicated in Hereditary Breast and/or Ovarian Cancer Syndrome Families"

_cancers, 2021, doi:10.3390/cancers13143406_

Round 1

Reviewer 1 Report

The article provides an interesting and thorough review of the special characteristics of the French Canadian (FC) population, the available data of rare variants in hereditary cancer associated and candidate genes and the advantages of analyzing these data to increase the knowledge of these gene associations.

From my point of view, the special features of FC population, genetically stratified into subpopulations, also involve an increased danger of using not fully matched control cohorts and finding spurious associations. I’ve missed some comments on this. This may be the case of RECQL, which several studies in admixed populations have failed to associate to hereditary breast cancer (HBC). In this sense the study by Dorling et al (PMID: 33471991) with 60,466 women with breast cancer and 53,461 controls is a notable contribution to the field that I think should enrich the discussion of the article.

Also the limitations of having many of the studies just genotyped already described variants should be discussed.

Cancer risk genes are presented in the article as they could have been described three or four years ago. Many of them with well stablished associations to breast or ovarian cancer risk (with definitive associations according to ClinGen, like PALB2, ATM, BRIP1 and CHEK2) are presented as just strong candidates. Others like BARD1 with known and definitive associations although controversial clinical utility are presented as new genes with unknown association. Indeed, in Scheme 2 "PALB2, RAD51D, and RECQL, new cancer predisposing gene candidates" can be read, throwing in the same pot such different situations.

I add here some minor comments:

  • Gene symbols should be written in italics throughout the text
  • Lines 108-109 “Depending on the population studied, between 5% to 20% of HBC and HBOC cancer 108 syndromes families have not been accounted for by PVs in BRCA1 and BRCA2 [5,31–34]. “ – I’d say that's the other way around: between 5 and 20% have been accounted for.
  • Figure 1: It is entitled “Representative carrier frequencies of pathogenic variants in HBC and HBOC predispos-126 ing genes in French Canadians of Quebec.” However, in some of the studies represented only one pathogenic variant (PV) was genotyped, instead of screening at least the coding sequence of these genes. I think that the variant and not just the gene should be written within the figure, not only in the footnotes.
  • Figure 2: It shows “The most frequently occurring pathogenic variants BRCA1, BRCA2, PALB2, and RAD51D 244 in French Canadians of Quebec and their allele frequency in other worldwide non-cancer popula-245 “ It would be very interesting the allele frequency in the FC non-cancer population. They/some seem to be available in citation 84. Since the FC population is so different from others, case-control aproximations should not use “as control” just worldwide populations.
  • Lines 281-284 “As shown in FCs, 84% 281 of BRCA1 and BRCA2 carrier positive HBC and HBOC syndrome families harbour one of 282 five specific PVs in these genes accounting for the high frequency of these PVs observed 283 in BC and OC affected individuals in this population [35]” - The study from citation 35 is performed by analysing just 20 previously described variants, not screening the genes. So this proportions are biased. It should be acknowledged.
  • Lines 311-314 – I’d say the numbers do not match between the text, figure 3 and Suppl Table 2.
  • Line 341 “...placing it among the least frequently occurring BRCA2 PVs in FC cancer families.” - I do not think that the authors should talk about the least frequently occurring variants in FC cancer population because many of the studies did not screen the gene but only looked for the previously described. The cited work did screen them, so I would refer just to this cohort and not compare it with the frequency of other minor variants in FC, which number and frequency is unknown.
  • Line 403 “PALB2 is the most promising of the newly proposed BC predisposing genes” - PALB2 is not promising, is a widely validated breast cancer predisposing gene, as asserted by ClinGen (https://search.clinicalgenome.org/kb/genes/HGNC:26144).
  • Line 428-430 “A targeted sequencing analysis of 54 candidate genes se-428 lected based on their function in HR repair in OC and controls by OCAC, only identified 429 a statistical association of potentially PVs in PALB2 with OC [112].” - Citation 112 found statistical association with other genes too.
  • Lines 480-481 “In cellulo assays showed that this variant encodes a protein isoform of RAD51D 480 Ser207Leu that affects the HR pathway function [119],” - Why do you call the mutated protein an isoform? this is misleading
  • Lines 485-486 “…and a variant of uncertain significance by 485 ACMG guidelines (Supplementary Table 3)” - No wonder that this variant is considered VUS by Varsome, an automatic resource that does not take into account case-control studies. Using Varsome as a proxy for ACMG classification (as you do according to the supplementary methods) is a clear limitation.
  • Line 498 “Less is known of the role of BRIP1 and CHEK2 cancer predisposing genes in confer-498 ring risk to BC and OC in FCs”- BRIP1 gene is definitively associated to OvCa and CHEK2 to BrCa (again, see ClinGen assertions).
  • According to Suppl Table 3, you use NM_001005735.2 as referene sequence for CHEK2. It would be more convenient to use NM_007194.4, the MANEselect transcript, which is also the fixed reference transcript in LRG. It gives place to a different cDNA position and codon (c.1217G>A, p.Arg406His), that would make the variant easier to find in ClinVar.
  • Lines 518-519 “c.1346G>A; p.Arg449His being the most promising candidate based on our in silico analysis” - I think that in silico analysis in missense variants is not a strong predictor of deleteriousness. Most labs submitting in ClinVar consider this variant likely benign, based in its high frequency in general populations, the small case-control study and a functional study. These arguments are stronger from my point of view.
  • The official MRE11A symbol is now MRE11 (https://www.genenames.org/data/gene-symbol-report/#!/hgnc_id/HGNC:7230).
  • Regarding the association of MRE11 to BC risk, maybe the article by Dorling et al (PMID: 33471991) supporting the same conclusions could be commented, as it works with big case and control sets.

Author Response

The article provides an interesting and thorough review of the special characteristics of the French Canadian (FC) population, the available data of rare variants in hereditary cancer associated and candidate genes and the advantages of analyzing these data to increase the knowledge of these gene associations.

From my point of view, the special features of FC population, genetically stratified into subpopulations, also involve an increased danger of using not fully matched control cohorts and finding spurious associations. I’ve missed some comments on this. This may be the case of RECQL, which several studies in admixed populations have failed to associate to hereditary breast cancer (HBC). In this sense the study by Dorling et al (PMID: 33471991) with 60,466 women with breast cancer and 53,461 controls is a notable contribution to the field that I think should enrich the discussion of the article.

Yes, we agree and for this reason included the latest efforts for obtaining FC population matched controls in the original version of the manuscript that was described in the last paragraph of the Perspective section.  We have also now addressed this issue further in the revised version regarding RECQL research, in line 816-818 (RECQL) by adding ‘This may be a result of not using fully matched control cohorts which may lead to spurious associations [48]’. We also added Dorling et al. citation (reference #48) to this statement which was referenced in the original version of the manuscript.

Also the limitations of having many of the studies just genotyped already described variants should be discussed.

Although we agree that many of the studies describe targeted variant analysis we have now clarified that some of these studies also include targeted gene sequencing analysis in lines 336-338 by adding ‘The ease of gene sequencing enabled the identification of new variants in the FC population using targeted gene sequencing of all exons and splice site regions’ in the revised manuscript.

Cancer risk genes are presented in the article as they could have been described three or four years ago. Many of them with well stablished associations to breast or ovarian cancer risk (with definitive associations according to ClinGen, like PALB2, ATM, BRIP1 and CHEK2) are presented as just strong candidates. Others like BARD1 with known and definitive associations although controversial clinical utility are presented as new genes with unknown association. Indeed, in Scheme 2 "PALB2, RAD51D, and RECQL, new cancer predisposing gene candidates" can be read, throwing in the same pot such different situations.

We wrote this review article keeping in mind that some of these new cancer predisposing genes have not been fully translated into clinical genetics settings. It’s for this reason that we used terminology “strong candidate” when describing these genes. However, we agree that this is being overly cautious and have changed the wording where appropriate.

Gene symbols should be written in italics throughout the text

Yes, this was a problem in submitting the original version of the manuscript and has been corrected.

Lines 108-109 “Depending on the population studied, between 5% to 20% of HBC and HBOC cancer 108 syndromes families have not been accounted for by PVs in BRCA1 and BRCA2 [5,31–34]. “ – I’d say that's the other way around: between 5 and 20% have been accounted for.

We’re referring to cancer syndrome families here and not sporadic cases. We’ve also corrected the 20% of 40% to represent all populations.

Figure 1: It is entitled “Representative carrier frequencies of pathogenic variants in HBC and HBOC predispos-126 ing genes in French Canadians of Quebec.” However, in some of the studies represented only one pathogenic variant (PV) was genotyped, instead of screening at least the coding sequence of these genes. I think that the variant and not just the gene should be written within the figure, not only in the footnotes.

We agree and changed the title in Figure 1 to ‘Representative carrier frequencies of frequently occurring pathogenic variants in HBC and HBOC predisposing genes in French Canadians of Quebec’ to reflect that the frequencies of the variants presented are largely specific variants found in this population. We tried to include more information in the Figure but it appeared cluttered and for this reason have retained the information regarding the specific variants in the figure legend, though we have added the variants for PALB2 and RAD51D to the figure.

Figure 2: It shows “The most frequently occurring pathogenic variants BRCA1BRCA2PALB2, and RAD51D 244 in French Canadians of Quebec and their allele frequency in other worldwide non-cancer popula-245 “ It would be very interesting the allele frequency in the FC non-cancer population. They/some seem to be available in citation 84. Since the FC population is so different from others, case-control aproximations should not use “as control” just worldwide populations.

While we agree that this would make for an interesting addition to this figure, the FC controls used in citation 84 differ from the population-based control data from gnomAD in that they were selected to not have a personal history of cancer or first-degree relatives with cancer unlike those from gnomAD. For this reason, we describe that data in the text. However, we have now clarified these FC controls in lines 305-307 ‘Indeed, a recent study has shown that BRCA1 and BRCA2 variants are rare (<0.2%) in the non-cancer FC population with no personal or family history of cancer relative to cancer cases [84].’

Lines 281-284 “As shown in FCs, 84% 281 of BRCA1 and BRCA2 carrier positive HBC and HBOC syndrome families harbour one of 282 five specific PVs in these genes accounting for the high frequency of these PVs observed 283 in BC and OC affected individuals in this population [35]” - The study from citation 35 is performed by analysing just 20 previously described variants, not screening the genes. So this proportions are biased. It should be acknowledged.

The nature of this study is now clarified in line 298-299 ‘As shown in an early targeted analysis of 20 variants in FCs, 84% of BRCA1and BRCA2 carrier positive HBC and HBOC syndrome families harbour one of five specific PVs in these genes accounting for the high frequency of these PVs observed in BC and OC affected individuals in this population’

Lines 311-314 – I’d say the numbers do not match between the text, figure 3 and Suppl Table 2.

The confusion is because we delineated the frequently occurring variants from those that have only been reported once in FCs. This is now clarified in line 335 by adding “only once” instead of “unique” to describe these variants as follows: “In reviewing the literature, 36 rare variants have been reported only once in BRCA1 and BRCA2 in FCs with BC or OC”

Line 341 “...placing it among the least frequently occurring BRCA2 PVs in FC cancer families.” - I do not think that the authors should talk about the least frequently occurring variants in FC cancer population because many of the studies did not screen the gene but only looked for the previously described. The cited work did screen them, so I would refer just to this cohort and not compare it with the frequency of other minor variants in FC, which number and frequency is unknown.

We agree and have now clarified that targeted gene sequencing analysis was performed in this study in line 373-377: ‘Although this variant has not been investigated to the same extent as other PVs in the FC population, targeted gene sequencing analysis identified BRCA2 c.9976A>T in two out of 256 (0.8%) unrelated HBC syndrome families [83], placing it among the least frequently occurring BRCA2 PVs in FC cancer families.’

Line 403 “PALB2 is the most promising of the newly proposed BC predisposing genes” - PALB2is not promising, is a widely validated breast cancer predisposing gene, as asserted by ClinGen (https://search.clinicalgenome.org/kb/genes/HGNC:26144).

This sentence was to introduce PALB2 as the most promising BC predisposing gene in a historical context based on previous discussions of other candidates. Note that in the original version of the manuscript, we ended the section by confirming that PALB2 is now among the high-risk BC predisposing genes in line 546.

Line 428-430 “A targeted sequencing analysis of 54 candidate genes se-428 lected based on their function in HR repair in OC and controls by OCAC, only identified 429 a statistical association of potentially PVs in PALB2 with OC [112].” - Citation 112 found statistical association with other genes too.

In this study by OCAC, though other candidates were identified with a statistical association with OC risk subsequent statistical analysis accounting for multiple testing (based on the Bayes false discovery probability) only revealed PALB2 as a strong candidate ovarian cancer susceptibility gene among the 54 candidate genes investigated.

Lines 480-481 “In cellulo assays showed that this variant encodes a protein isoform of RAD51D 480 Ser207Leu that affects the HR pathway function [119],” - Why do you call the mutated protein an isoform? this is misleading

We understand how this terminology could be confusing to readers and have changed protein isoform to aberrant protein. This has been clarified in lines 617-619 ‘In cellulo assays showed that this variant encodes an aberrant protein, RAD51D p.Ser207Leu, that affects the HR pathway function [119], and thus may be pathogenic and play a role in conferring risk to OC in carriers.’

Lines 485-486 “…and a variant of uncertain significance by 485 ACMG guidelines (Supplementary Table 3)” - No wonder that this variant is considered VUS by Varsome, an automatic resource that does not take into account case-control studies. Using Varsome as a proxy for ACMG classification (as you do according to the supplementary methods) is a clear limitation.

We are not clear on this comment. We have not used Varsome as a proxy for ACMG classification of variants in this report, but have used it to readily obtain ACMG classification of variants, as Varsome does not determine classifications but reports classifications from other sources (for example from ACMG, ClinVar, gnomAD). However, to avoid this misunderstanding, we clarified how we used this resource in the supplementary methods by changing “determine” to “extract” as follows:  “Varsome (www.varsome.com) [17] was used to extract the American College of Medical Genetics and Genomics (ACMG) classification (pathogenic, likely pathogenic, uncertain significance, likely benign, benign) and was last accessed on April 2021.”

Line 498 “Less is known of the role of BRIP1 and CHEK2 cancer predisposing genes in confer-498 ring risk to BC and OC in FCs”- BRIP1 gene is definitively associated to OvCa and CHEK2 to BrCa (again, see ClinGen assertions).

We have now clarified this statement by changing line 635 to read “Relative to BRCA1 and BRCA2, less is known of the role of BRIP1 and CHEK2 cancer predisposing genes in conferring risk to BC and OC in FCs”

According to Suppl Table 3, you use NM_001005735.2 as referene sequence for CHEK2. It would be more convenient to use NM_007194.4, the MANEselect transcript, which is also the fixed reference transcript in LRG. It gives place to a different cDNA position and codon (c.1217G>A, p.Arg406His), that would make the variant easier to find in ClinVar.

We agree and this has been edited in the tables in the text to reflect transcript NM_007194.4

Lines 518-519 “c.1346G>A; p.Arg449His being the most promising candidate based on our in silico analysis” - I think that in silico analysis in missense variants is not a strong predictor of deleteriousness. Most labs submitting in ClinVar consider this variant likely benign, based in its high frequency in general populations, the small case-control study and a functional study. These arguments are stronger from my point of view.

While this variant may be considered “likely benign”, the selected in silico tools have an over 90% positive predictive value with variants in ClinVar whether they are pathogenic, VUS, or benign. The study (Ghosh et al. citation 30) assessed a number of in silico tools against ClinVar variants for accuracy and precision and from this list we selected the in silico tools that had accuracy and precision of over 90%.

The official MRE11A symbol is now MRE11 (https://www.genenames.org/data/gene-symbol-report/#!/hgnc_id/HGNC:7230).

Yes (thank you), and this has been edited throughout the revised manuscript.

Regarding the association of MRE11 to BC risk, maybe the article by Dorling et al (PMID: 33471991) supporting the same conclusions could be commented, as it works with big case and control sets.

We have already referenced the Dorling et al study but have now revised line 713-716 to highlight the similar finding as follows: ‘However, a recent study of MRE11 c.1516G>T; p.Glu506Ter, which has been reported in multiple FC cancer cases and also found in other populations, suggested that it may not to be associated with BC risk [136], which is in line with recent findings from a large BC case-control study [48]’

Reviewer 2 Report

Authors comprehensively performed an in-depth description of germline genetic studies in hereditary breast and/or ovarian cancer patients from Quebec. 

I would make the following general suggestions:

  1. Consider re-structuring the abstract. Although there is no need to use subtitles within the abstract, I would recommend writing them down and following a structure.
  2. Consider reducing the number of words, adding more subtitles to the whole article, avoiding personal opinions and stating the facts. It is an extensive work that I think could be of much help for the scientific community but right now is a very dense reading material and it can be difficult to find the important information within such a wide story.
  3. I suppose it was a problem at the time of submitting, but all genes should be in italics.

Hereby some detailed content remarks by line:

  1. Consider adding the specification of “review article” to the title according to PRISMA guidelines.
  2. Line 48 and 61: Although you are telling that BRCA involvement in BC/OC is major in line 48 and that BRCAs implication in HBOC is major in line 61, I would suggest avoiding the first statement. You could say that it is the most prevalent gene implicated in both sporadic and hereditary breast and OC, trying not to sound redundant.
  3. Also to avoid redundancy, could you maybe structure the introduction something like “Genetic screening in BC/OC” (lines 46..), “Discovery of BRCAs” (lines 66-77)“ , etc.
  4. Line 93: after talking about the Icelandic population continue in a new paragraph
  5. Consider reducing the word count especially in “2.Methods….” to at least the half to lighten the read.
  6. Line 104: delete the word “cancer” after OC.
  7. The information from the literature in the footnote of Figure 1 is useful but difficult to digest as it is. I would suggest adding all the information on superscripts in a table within the same figure (in panels).
  8. Line 159: It is states that 10 new cancer related genes have been proposed to be associated to HBC/HBOC, and refer to supplementary table 1. In ST1 you listed 14 genes including BRCAs, which would be 12 genes?
  9. I feel the lines 174-175 and 182 and 183 redundant.
  10. Lines 182-197 are not FC related, should be elsewhere.
  11. In figure 2 you are saying that those are the most frequent variants in FC. Is there a way to add the FC-allele frequency to the figure?
  12. Line 282: One of the two words (carriers/positive) can be deleted.
  13. Lines 284-288: This statement has low relevance if the FC carrier frequency is not stated.
  14. Lines 297-299: Find redundant to say that there are 18 PVs being the majority nonsense/frameshift resulting in LoF and in lines 299-301 say that there are rare in non-cancer supporting an association with cancer.
  15. Figure 3: Please add the coding DNA change to all variants. The most frequently occurring are differentiated already by the third dotted line.
  16. Figure 3 footnote: Since you already comment in the text where the classification is coming from, there is no need to add it to the footnote. I would recommend deleting the 2nd and 3rd
  17. Line 317: Delete the word “variants” at the end of the sentence.
  18. Line 324: Are you positive about this? I would say there are at least a couple of papers nicely comparing in silico and in vitro results from BRCA variants.
  19. Lines 329-330: Consider rephrasing.
  20. Line 350: What do you mean with “has not been observed in FC”? Can you please add the reference.
  21. Lines 351-352: I do not see how this relates to the rest of the paragraph. Maybe would fit better in the next.
  22. Line 355: Consider mention the information on BRCA14327C>T when first talking about this variant and common ancestors and not in the spectrum.
  23. Lines 358-361: I do not feel there is coherence with the rest of the paragraph.
  24. Lines 365-368: I would recommend avoiding saying you are going to comment on something later. It only makes it longer to read.
  25. Line 379: Comparing hereditary cancer affected and overall population (cancer and non-cancer) seems odd to me.
  26. Line 379: Figure 1f says 3.8% and here 1.2%, could you explain why?
  27. Lines 384-386: Consider rephrasing to something like: only one rare missense variant (c.1062….) in STK11 was identified in 96 BRCA negative FC families with HBC/HBOC and no Peutz-Jaghers phenotype.
  28. Line 410: Please add how many PALB2 carriers of how many studied genotypes.
  29. Line 413: “Truncated” aberrant” seem redundant to me.
  30. Line 420: Since PALB2 is the first example, why not consider talking about PALB2 before TP53?
  31. Line 430-433 and 439-443: Consider moving these general facts at the beginning of this topic.
  32. Line 468: Consider changing “rare carriers” to carriers of rare variants.
  33. Line 548: Consider rephrasing.
  34. Line 599: I am missing a preposition.
  35. Lines 605-606: Consider relocating to point 4.4

Author Response

Authors comprehensively performed an in-depth description of germline genetic studies in hereditary breast and/or ovarian cancer patients from Quebec. 

I would make the following general suggestions:

1. Consider re-structuring the abstract. Although there is no need to use subtitles within the abstract, I would recommend writing them down and following a structure.

We acknowledge this reviewer’s comments but felt that we had succeeded in doing just as suggested. Although there are no specific guidelines for abstracts for review articles for this journal, we have: 1) presented sufficient background on the subject of this article; 2) what we proposed to discuss in this review article regarding the merits of genetically characterizing cancer predisposing genes in FCs of Quebec, with our focus on hereditary breast and ovarian cancer syndrome families; 3) we specifically outline in three general topics of discussion on how we make our argument that the study of FCs facilitates the characterization of genetic variants in this context; and 4) completed the abstract with a concluding statement. We defer to the editor for further instructions.

2. Consider reducing the number of words, adding more subtitles to the whole article, avoiding personal opinions and stating the facts. It is an extensive work that I think could be of much help for the scientific community but right now is a very dense reading material and it can be difficult to find the important information within such a wide story.

Our intention with this review article was to present sufficient background for those less familiar with the discovery and validation of hereditary breast and ovarian cancer predisposing genes to lay the groundwork for appreciating the current challenges in identifying new genes. Included is background information regarding the discoveries of BRCA1 and BRCA2 (seminal discovery of BRCA1 in scheme 1) and details concerning the more recent discoveries of newer genes, such as PALB2 and RAD51D to contrast past and present methodologies.  We also used schemes to provide additional information for those less familiar with history of FCs of Quebec to more fully appreciate genetic drift in this unique population. Perhaps in doing so this created a lengthier article for those who are familiar with these topics but aim in preparing this comprehensive review is also a broader audience. We also presented the topics in a way to reflect publications of FC studies as they occurred in the literature, and thus think it is important to keep the narrative intact as presented.

We are unaware of any personal opinions that were stated in this review article. It would have been helpful if examples were provided for us to review these statements as our intention is not to come across as opiniated on matters that not supported by facts.

3. I suppose it was a problem at the time of submitting, but all genes should be in italics.

Yes, this was a problem in submitting the original version of the manuscript and has been corrected.

Hereby some detailed content remarks by line:

1. Consider adding the specification of “review article” to the title according to PRISMA guidelines.

This article is a review article and notation has been corrected on the first page of the article. However, as this article is a comprehensive review of focused topic that is still evolving in the field of hereditary cancer predisposing genes and not a systematic review, we have not followed the PRISMA guidelines. We envision, a systematic review according to PRISMA guidelines would be possible in the future as more research is conducted on the genetic architecture of FC population of Quebec (not only on the subject of hereditary cancer families and cancer predisposing genes) and published.

2. Line 48 and 61: Although you are telling that BRCA involvement in BC/OC is major in line 48 and that BRCAs implication in HBOC is major in line 61, I would suggest avoiding the first statement. You could say that it is the most prevalent gene implicated in both sporadic and hereditary breast and OC, trying not to sound redundant.

We agree and have now removed ‘major’ from line 48.

3. Also to avoid redundancy, could you maybe structure the introduction something like “Genetic screening in BC/OC” (lines 46..), “Discovery of BRCAs” (lines 66-77)“ , etc.

We respectfully disagree and think that adding additional subsections in the introduction would break the continuity of the introduction as each of these paragraphs introduce screening and discovery of BC/OC genes and are not comprehensive in and of themselves.

4. Line 93: after talking about the Icelandic population continue in a new paragraph

Yes, we agree. This was likely an error in submission.

5. Consider reducing the word count especially in “2.Methods….” to at least the half to lighten the read.

We defer to comments made above regarding reducing the text.

6. Line 104: delete the word “cancer” after OC.

Yes, this is now deleted.

7. The information from the literature in the footnote of Figure 1 is useful but difficult to digest as it is. I would suggest adding all the information on superscripts in a table within the same figure (in panels).

Our goal with this figure is to provide as accurate as possible frequencies of specific pathogenic variants in defined cancer groups in FCs of Quebec. As these studies were done over a period of time and with different study groups it was not possible to integrate the data into one figure for each category. We placed the detailed information in the figure and found it cluttered and what we thought was a loss of information about proportions, which is why we included this information in the figure legend.

8. Line 159: It is states that 10 new cancer related genes have been proposed to be associated to HBC/HBOC, and refer to supplementary table 1. In ST1 you listed 14 genes including BRCAs, which would be 12 genes?

Yes, you are correct (thank you). This has been edited to reflect the 12 genes described in the supplementary table.

9. I feel the lines 174-175 and 182 and 183 redundant.

Lines 174-175 are referring to next-generation sequencing (whole exome or whole genome) and multi-gene panels and lines 182-183 are referring to targeted gene sequencing of individual genes. For clarification, “targeted” has been added to line 193: “With the identification of BRCA1 and BRCA2, their role in conferring risk to BC and OC in various populations were investigated by targeted gene sequencing analyses of cancer cases.”

10. Lines 182-197 are not FC related, should be elsewhere.

We have retained this statement, as we think it necessary to provide the background information as an introductory section which sets the stage for discussing variants in the context of FCs.

11. In figure 2 you are saying that those are the most frequent variants in FC. Is there a way to add the FC-allele frequency to the figure?

While we agree that this would make for an interesting figure, the FC controls used in citation 84 differ from the population-based control data from gnomAD in that they were selected to not have a personal history of cancer or first-degree relatives with cancer unlike those from gnomAD. For this reason, we describe that data in the text. The nature of the controls are now clarified in lines 305-307 ‘Indeed, a recent study has shown that BRCA1 and BRCA2 variants are rare (<0.2%) in the non-cancer FC population with no personal or family history of cancer relative to cancer cases [84].’

12. Line 282: One of the two words (carriers/positive) can be deleted.

Carrier has been removed.

13. Lines 284-288: This statement has low relevance if the FC carrier frequency is not stated.

Yes, we agree. We have now clarified the statement by adding the following in lines 305-307 ‘Indeed, a recent study has shown that BRCA1 and BRCA2 variants are rare (<0.2%) in the non-cancer FC population with no personal or family history of cancer relative to cancer cases [84].’ Which is taken in part from a statement that was removed from the end of the subsequent paragraph in Section 3.2.

14. Lines 297-299: Find redundant to say that there are 18 PVs being the majority nonsense/frameshift resulting in LoF and in lines 299-301 say that there are rare in non-cancer supporting an association with cancer.

We agree and have edited that sentence. Also, view our response to item 13 above.

15. Figure 3: Please add the coding DNA change to all variants. The most frequently occurring are differentiated already by the third dotted line.

Yes, we agree, the HGVS nomenclature has been added for all variants.

16. Figure 3 footnote: Since you already comment in the text where the classification is coming from, there is no need to add it to the footnote. I would recommend deleting the 2nd and 3rd

Yes, this has been removed from the footnote. The figure legend has been revised.

17. Line 317: Delete the word “variants” at the end of the sentence.

Yes, removed.

18. Line 324: Are you positive about this? I would say there are at least a couple of papers nicely comparing in silico and in vitro results from BRCA variants.

Though we agree with this statement, the papers we are aware of compared in silico and in vitro results in studies of BRCA1 and BRCA2 that were not systematic for all possible variants but were focused on specific regions within the genes. We are not aware of any systematic study of splicing in silico tools conducted in the same way as for those in silico tools that predict effects on protein function as described in Ghosh et al. (citation 30 in the manuscript).

19. Lines 329-330: Consider rephrasing.

For clarification, we rephrased the sentence in lines 354-365 “Of note, two of these missense variants, BRCA1 c.736T>G; p.Leu246Val and BRCA2 c.8850G>T; p.Lys2950Asn, did not affect the function of the HR pathway [91,92]’

20. Line 350: What do you mean with “has not been observed in FC”? Can you please add the reference.

We clarified this statement as follows in line 385-387 “However, this has not been studied in FCs due to the overall low frequency of carriers in this population enabling statistical associations of each variant with risk of BC or OC (Figure 3).”

21. Lines 351-352: I do not see how this relates to the rest of the paragraph. Maybe would fit better in the next.

We agree and we have moved this statement to the subsequent paragraph.

22. Line 355: Consider mention the information on BRCA14327C>T when first talking about this variant and common ancestors and not in the spectrum.

The rationale for including discussion of this variant in this paragraph is to emphasize that carriers of this variant have traced their ancestors to different regions across Quebec where carriers of other variants were traced to specific regions within the province.

23. Lines 358-361: I do not feel there is coherence with the rest of the paragraph.

This has been separated into a new paragraph which introduces the next section.

24. Lines 365-368: I would recommend avoiding saying you are going to comment on something later. It only makes it longer to read.

As this is a comprehensive review where there were only a few studies of FC of other established genes, such as TP53 and STK11, we felt that this was the best place to include this information prior to describing the candidate cancer predisposing genes associated with HBC and HBOC syndromes.

25. Line 379: Comparing hereditary cancer affected and overall population (cancer and non-cancer) seems odd to me.

This sentence was to emphasize the rarity of identifiable TP53 carrier families in the general population and as a consequence the rarity of TP53 carriers in the general population.

26. Line 379: Figure 1f says 3.8% and here 1.2%, could you explain why?

Thank you for pointing out this discrepancy, but a matter of not including all available information in the final version of the original manuscript. The data from Figure 1 refers to the percentage of TP53-positive HBC families which is 3.8% and the 1.2% frequency mentioned in the text of the manuscript refers to the carrier frequency of sporadic BC cases. We have clarified the statement by adding this information in line 417 ‘The overall estimated carrier frequency of PVs in TP53 in HBC families at 3.8% [102] (Figure 1f) and 1.2% in sporadic BC cases [103] was higher than expected given the estimated 1 in 5,000 to 20,000 TP53-carriers in the general population worldwide (re-viewed in [104]).’

27. Lines 384-386: Consider rephrasing to something like: only one rare missense variant (c.1062….) in STK11 was identified in 96 BRCA negative FC families with HBC/HBOC and no Peutz-Jaghers phenotype.

This sentence has been clarified as follows in lines 461-464: ‘Only one rare variant in STK11 was identified in a study of 96 BRCA-negative HBC families, where the carrier family that did not exhibit clinical features consistent with Peutz-Jeghers syndrome [105] (Supplementary Table 4).’

28. Line 410: Please add how many PALB2 carriers of how many studied genotypes.

We have now clarified this section by revising the following sentences lines 512-517 “Soon thereafter, targeted sequencing analysis of PALB2 in 50 FC early-onset or familial BC cases identified one carrier of PALB2 (NM_001005735.2): c.2323C>T; p.Gln775Ter and this variant was also identified in 2/356 BC cases but not in controls [108]. Carriers were subsequently identified in 2% of FC HBC families BRCA-negative for the five most commonly occurring PVs observed in FCs (Table 2, Figure 1d, Figure 2).’ The following sentence was incorporated into the above sentences “The same PALB2 variant was found in several FC BC cases but not in cancer-free controls [108]’

29. Line 413: “Truncated” aberrant” seem redundant to me.

We agree, “aberrant” removed.

30. Line 420: Since PALB2 is the first example, why not consider talking about PALB2 before TP53?

We respectfully disagree since TP53 has been recognized as a cancer predisposing gene since 1990. There were reports of investigating the possibility that TP53 involvement in some BRCA1/2 negative HBC syndrome families given the young age of onset of BC and frequency of the BC in families that may have not initially identified as Li Fraumeni syndrome families in their clinical presentation. Indeed, this was the rationale for studying TP53 in FC HBC families.

31. Line 430-433 and 439-443: Consider moving these general facts at the beginning of this topic.

These statements were made to reflect the timeline in which the studies referring to them were published, which occurred in the context of studying PALB2 in the FC population.

32. Line 468: Consider changing “rare carriers” to carriers of rare variants.

Removed “rare” from this sentence.

33. Line 548: Consider rephrasing.

This has already been rephrased considering the first reviewers comments “However, a recent study of MRE11 c.1516G>T; p.Glu506Ter, which has been reported in multiple FC cancer cases and also found in other populations, suggested that it may not to be associated with BC risk [136] similar to recent findings in a large BC case-control study [48]”

34. Line 599: I am missing a preposition.

Added the word “of”.

35. Lines 605-606: Consider relocating to point 4.4

We respectfully disagree because everything described up to this point is work involving established/proposed new cancer predisposing genes that have been studied in the FC population in the context of HBC and/or HBOC. This statement is placed in the perspectives section to highlight the fact that while other cancer predisposing genes have been reported in the literature there are no studies (we are aware of) that have investigated these specific genes (ie FANCC, FANCM and RAD50)  in the FCs.